# Network Approaches to Study Endogenous RNA Competition and Its Impact on Tissue-Specific microRNA Functions

**DOI:** 10.3390/biom12020332

**Published:** 2022-02-19

**Authors:** Tânia Monteiro Marques, Margarida Gama-Carvalho

**Affiliations:** BioISI—Biosystems & Integrative Sciences Institute, Faculty of Sciences, University of Lisboa, 1749-016 Lisboa, Portugal; trmarques@fc.ul.pt

**Keywords:** microRNAs, post-transcriptional regulation, ceRNA networks

## Abstract

microRNAs are small non-coding RNAs that play a key role in regulating gene expression. These molecules exert their function through sequence complementarity with microRNA responsive elements and are typically located in the 3′ untranslated region of mRNAs, negatively regulating expression. Even though the relevant role of miRNA-dependent regulation is broadly recognized, the principles governing their ability to lead to specific functional outcomes in distinct cell types are still not well understood. In recent years, an intriguing hypothesis proposed that miRNA-responsive elements act as communication links between different RNA species, making the investigation of microRNA function even more complex than previously thought. The competing endogenous RNA hypothesis suggests the presence of a new level of regulation, whereby a specific RNA transcript can indirectly influence the abundance of other transcripts by limiting the availability of a common miRNA, acting as a “molecular sponge”. Since this idea has been proposed, several studies have tried to pinpoint the interaction networks that have been established between different RNA species and whether they contribute to normal cell function and disease. The focus of this review is to highlight recent developments and achievements made towards the process of characterizing competing endogenous RNA networks and their role in cellular function.

## 1. microRNAs as Post-Transcriptional Regulators of Gene Expression

MicroRNAs (miRNAs) are a class of evolutionarily conserved small non-coding (nc) RNAs that are approximately 18–22nt long and that were first identified in *Caenorhabditis elegans* in 1993 [1,2]. Since their discovery, these small RNA molecules have been studied extensively and are currently known for their role in gene expression regulation, acting post-transcriptionally to repress protein translation or to induce messenger RNA (mRNA) degradation [3]. To date, there are 576 bona fide miRNA genes that have been identified in the human genome [4], although the actual number may be up to three times higher [4,5].

miRNAs can originate from intergenic regions of the genome, meaning that their transcription is independent of host gene and intragenic regions, mostly coming from introns of protein coding genes and thus sharing the same promoters [3]. The canonical biogenesis pathway of miRNAs starts with the transcription of a primary microRNA (pri-miRNA) by RNA polymerase II in the nucleus. These transcripts are then processed by Drosha, which cuts them into 70 to 100nt hairpin-like transcripts called precursor miRNA (pre-miRNAs). Exportin-5 (XPO5) exports the pre-miRNAs from the nucleus to the cytoplasm, where the RNAse III endonuclease Dicer cleaves the transcripts into miRNA:miRNA* duplexes by removing the terminal loop [6]. Generally, one strand of this duplex is loaded into a ribonucleoprotein complex called the RNA-induced silencing complex (RISC) by an Argonaute protein (Ago), while the other strand, known as miRNA star (miRNA*), is degraded. In some instances, however, both strands can function as mature miRs and will be distinguished by the suffix -5p or -3p, depending on their relative position in the pre-miR hairpin [3]. Once loaded into the RISC complex, the miRNA can bind to an miRNA-responsive element (MRE) that is typically located in the 3′ untranslated region (UTR) of the target mRNA. This binding is primarily determined by the complementary hybridization of the so-called “seed sequence”, corresponding to nucleotides 2 to 7 of the miRNA [7]. This binding leads to a negative regulatory effect on the mRNA molecule. The type of post-transcriptional repression that is exerted by an miRNA is partly dictated by whether the complementary binding to the 3′UTR of the target mRNA is perfect or not: if the miRNA binds with perfect complementarity to the target mRNA, then RISC cleaves it. If the base pairing is partial, RISC cleavage is prevented, and additional proteins are recruited to induce translational repression and/or 3′-end mRNA deadenylation [7,8]. Unlike plants, in which miRNAs predominantly display full complementarity to the 3′ UTR MREs, leading to mRNA cleavage, in animals, there is close and complex interplay between miRNA-dependent translational inhibition and mRNA degradation. Although the most dominant effect of miRNAs on their targets is associated with mRNA decay, regulatory interactions involving reversible translational inhibition have been described in oocytes, early embryos, and neurons [7,9].

## 2. microRNA Regulatory Networks Are Central Coordinators of Cellular States

The discovery that miRNAs were the negative regulators of the *C. elegans* heterochronic gene lin-14 immediately spotlighted their mode of action as being linked to partial 3′UTR anti-sense complementarity [1,2] and revealed that miRNAs were integrated in the regulatory pathways controlling the temporal gene expression patterns underlying developmental timing [10]. Early on, these regulatory pathways were shown to rely on a complex regulatory network involving miRNAs, transcription factors (TFs), RNA binding proteins (RBPs), and other cell fate regulators, establishing feed-back regulatory loops [11]. In the following years, the discovery of hundreds of miRNA genes in plants and animals, including the discovery of the orthologues of these *C. elegans* miRNAs, suggested that such regulatory networks were likely to play a central role in the coordination of cellular processes. The first step towards their characterization was obviously the systematic identification of the target mRNAs, for which no efficient experimental methods were available. Given the understanding of the basic mechanisms governing miRNA function, several computational methods for target prediction were developed and progressively refined [12,13,14,15,16,17]. Prediction algorithm features included the complementarity of the seed sequence to the target mRNA; the calculated free energy of the interaction; stability characterization; determining whether the interacting sequences were evolutionary conserved; and the target site number, location, and accessibility (reviewed by [18]). These early studies focusing on multiple species revealed two common themes: (1) a given miRNA could target tens to hundreds of mRNA molecules, many of which were involved in the same cellular pathways, and (2) a given mRNA species could be targeted by different miRNAs. Correlation analysis of miRNA and mRNA target expression supported the view that miRNAs have a widespread impact on mRNA repression and, quite strikingly, on target sequence evolution. Indeed, the avoidance of selected MREs in the transcript sequences that are expressed in the same tissues as a given miRNA is observed consistently [19].

In the absence of experimental approaches to perform a large-scale mapping of miRNA–mRNA interactions, target prediction algorithms supported the creation of the first RNA–RNA interaction (RRI) databases [18]. In addition to facilitating the characterization of the cellular and developmental roles of specific miRNAs by providing tools to identify potential mRNA targets and interaction networks (see, for example, [20]), these tools supported the development of network-based approaches to explore the global principles of miRNA-dependent regulation. By integrating miRNA–mRNA interaction maps with gene expression data, it was possible to identify the existence of specific modes of regulation, akin to previous work on TF regulatory networks [21]. Of note, these studies revealed how the same miRNA may be simultaneously involved in different types of regulatory networks depending on the specific target considered and that very often, miRNA networks include TFs, particularly those that regulate them directly (reviewed by [22]).

## 3. Challenges in Predicting Functional Outcomes of microRNA Regulatory Networks

Mapping the networks formed by miRNA–mRNA interactions is critical for identifying the cellular functions regulated by these molecules. Advances in experimental methods currently support the large-scale profiling of these interactions. Together with the development of databases compiling experimentally validated interactions derived from publications, this has significantly improved the available information on miRNA targets, serving as a complement to computational predictions [23]. Strikingly, miRNA–mRNA networks may be less informative regarding miRNA function than the use of protein–protein interaction (PPI) networks for targets. PPI networks can be used to identify connections between genes showing altered expression in response to changing miRNA levels and thus hint at cellular outcomes. An example of the successful use of a network-based approach to characterize the specific function of an miRNA is the identification of miR-24 as a cell proliferation inhibitor [24]. Following typical approaches in the field, steady-state mRNA levels were quantified in cells over-expressing miR-24, revealing a universe of ~200 down-regulated genes, of which approximately half were predicted direct targets. These numbers are consistent with other published work and reflect both the existence of indirect miRNA targets and the lack of sensitivity in prediction algorithms [23]. The responsive genes formed a tight PPI network that included E2F2—for which no MRE was identified—as a central hub. Experimental assays validated E2F2 as a direct target of miR-24 and its downregulation as the mechanism behind the observed anti-proliferative phenotype.

Although many studies correlating gene expression changes with miRNA interaction networks reveal a significant correlation between the presence of one or more MREs and transcript down-regulation, the high number of upregulated genes that are also predicted targets is generally overlooked. Indeed, several comparative analyses of prediction methods highlight the high level of false-positive results as well as the lack of concordance between different algorithms [25]. To make the task of understanding the cellular functions of miRNAs even more daunting, the co-regulation of mRNAs encoding products with antagonistic functions is often observed, as reported, for example, in our previous work on miR-34c-5p [26]. Even though one could expect that the current understanding of the rules governing miRNA function should be able to support robust predictions of their cellular role, this type of contradictory scenario is not uncommon. For example, miRNAs are often classified as being oncogenes (oncomiRs) or tumor suppressors, depending on the observed functional outcome of their over-expression or repression on cell proliferation and survival. The observations establishing these associations were made in the first decade after miRNA discovery [27], suggesting that it would be possible to define a global function for an miRNA. However, conflicting reports regarding oncogenic or tumor-suppressive roles quickly started to appear in the literature [28]. Thus, a single miRNA could be described as being a tumor suppressor in a given cancer and oncogenic in another. For example, miR-125b has been identified as being oncogenic in most hematological contexts but is a tumor suppressor in solid tumors as well as in chronic lymphocytic leukemia [29]. Another example is the well-known “oncomiR” miR-21, which has been shown to promote cell proliferation in ovarian cancer [30], hepatocellular carcinoma [31], and melanoma [32] but was described as an tumor progression inhibitor in chondrosarcoma [33]. Similarly, miR-186 has been characterized as being up- and down-regulated in different types of cancer, acting both as an oncomiR and a tumor suppressor [34]. A systematic review of the reported contradictory differences in miR-186 expression suggested potential confounding factors emerging from the number of samples, heterogeneity, and classification. However, the authors highlight that variation in target abundance or in dose-dependent effects might play an important role in determining whether an miRNA will have an oncogenic or tumor suppressive effect [34]. Looking at the targets of miR-125b, Svoronos and colleagues pointed to the fact that it can regulate both oncogenes and tumor suppressors and proposed that the outcome on a specific cell type would be influenced by the relative expression levels of the targets [28]. This hypothesis was recently confirmed in a study that explored the PPI network of the pro-apoptotic and anti-apoptotic targets of miR-125b using a mathematical model to determine the impact various target abundances would have on the apoptotic pathway [30]. The results were validated experimentally, showing that it was possible to reverse the effect of miR-125b from being pro-apoptotic to being anti-apoptotic through prior reductions in the levels of two of its three pro-apoptotic targets (MCL and BCL-w). Interestingly, the knock-down of the third target (BCL-2) did not alter the pro-apoptotic phenotype, with the authors suggesting that this was a consequence of its lower expression levels in the cell line used in their study [30].

Previously, the concept that the concentration of target mRNA molecules directly influences miRNA activity had been proposed [35]. Analyzing mRNA expression changes in response to the overexpression of miRNAs in two different cell lines, the authors showed that target down-regulation was a function of target abundance. This led to the proposal that a more precise and, if possible, quantitative definition of a gene as miRNA target or non-target should be implemented [35]. This problem is compounded by the fact that current target prediction algorithms are blind to the context where the miRNA is being expressed, not only regarding the abundance of the targets, but also of the actual sequence isoforms that are present. Indeed, extensive transcriptome profiling studies have revealed that the use of alternative 3′UTRs is widespread across different cell types and will determine the presence of MREs [36]. The availability of deep sequencing data has also allowed for a deeper understanding of the diversity of microRNA sequences, revealing that miRNAs can have multiple variants, known as isomiRs [37]. These sequences, once considered artifacts, are now known to be functional mRNA-interacting molecules and represent variations in the length and sequence composition of the same miRNA locus. IsomiRs can be catalogued into three classes: 3′ isomiRs, 5′ isomiRs, and polymorphic isomiRs. The 3′ isomiRs are usually less likely to affect target interaction, while the 5′ isomiRs, which have an altered seed region, can have different target affinities compared to the reference sequence. These sequence variations can happen in disease- or tissue-specific context. Of note, the reference sequence annotated in the miRBase repository [38], which is the canonical sequence, i.e., the sequence that is used as reference for all of the algorithms that predict miRNA–mRNA interactions, does not always correspond to the most abundant miR isoform. Indeed, a recent systematic evaluation of the miRbase repository revealed that on average, the most abundant sequences are one nucleotide shorter than the reference sequence [39]. To our knowledge, the actual impact of these discrepancies on miRNA target prediction has not been systematically assessed.

In summary, even though the relevant role of miRNA-dependent regulation is broadly recognized, predicting the specific functional outcome of miRNA expression in distinct cell types is still a daunting task. The inhibitory role of miRNAs in gene expression does not imply that their function in regulatory networks is simply repressive. In fact, each individual miRNA is extremely dependent on the unique context in which the interactions occur. Mapping miRNA–mRNA interactions—either through computational prediction or experimental approaches—and building and analyzing regulatory networks incorporating gene expression data while often integrating RRI, PPI, and TF interactions has provided important insights into miRNA functions during cellular regulation. However, the development of approaches that are able to successfully predict the functional outcomes of the presence of a given miRNA in a system has remained extremely limited. This is likely due to the fact that in addition to the limited quality of the interaction data, many relevant aspects of the system, including the sequence isoforms and abundance, are not usually considered.

## 4. The ceRNA Hypothesis—A Transcriptome Wide Intrinsic Regulatory Network

As described above, the early days of research into miRNA function revealed that these molecules establish complex regulatory networks that can encompass both radical switches to cellular and developmental programs, and more subtly, non-essential buffering activities to reduce the noise during gene expression. In the last decade, several lines of evidence have led to significant changes in perspective regarding the nature and functional outcomes of RRIs and their associated regulatory networks. To address the paradox of why only a small number of mRNAs with identified MREs show significant changes in expression in response to miRNA levels, an alternative hypothesis was proposed [18]. This hypothesis suggests that most targets are in fact competitive inhibitors of miRNAs, sequestering them and thus indirectly modulating the levels of the “true” targets [18]. This model of competitive regulation started gaining traction after the identification of new species of non-coding RNAs that was shown to exert regulatory functions by sequestering miRNAs. The molecules that were involved in this process included long non-coding RNAs (lncRNA), circular RNAs (circRNA), and pseudogenes, which are often referred to as “miRNA sponges”. The term “sponge” was used by Ebert and colleagues [40] when referring to decoy RNA targets for miRNAs that act as competitive inhibitors of their function.

Pseudogenes are duplicated versions of parental genes or, most often, retrotransposed transcripts (processed pseudogenes), that no longer have the ability to produce protein products. A pioneer study carried out by Poliseno et al. [41] revealed that the tumor suppressor PTEN and its non-coding pseudogene PTENP1 established a co-regulatory interaction, whereby the 3′UTR of PTENP1 serves as a decoy for the miRNAs acting on the PTEN mRNA. To increase the complexity of this interaction, a later study showed that the PTENP1 locus could produce and antisense lncRNA, which regulates the PTENP1 sense transcript through direct base-pairing interactions involving the first exon sequence [42]. lncRNAs are defined as RNAs that are longer than 200 nucleotides that do not encode any functional protein [43]. This definition encompasses a highly heterogeneous group of transcripts regarding both biogenesis and functions. Among the latter, the regulation of gene expression by sponging miRNAs was first demonstrated in 2011 for linc-MD1 [44]. This lncRNA was found to act as decoy for two miRNAs, miR-133 and miR-135, controlling the timing of myoblast differentiation [44]. Around the same time as these discoveries, the universe of non-coding RNAs was being expanded by the identification of circular RNA molecules. circRNAs are a class of ncRNA molecules that are derived from linear mRNA precursors that undergo a back-splicing process on one or more exons, forming a closed-loop structure [45]. In the same issue of *Nature*, Hansen et al. [45] and Memczak et al. [46] reported the identification of a conserved circRNA (CDR1 as or ciRS-7) containing 70 MREs for miR-7. This molecule acts as a decoy to reduce miR-7 availability in cells while remaining unaffected by miR binding. Interestingly, prior work by Hansen and co-workers had revealed that this circRNA is a single perfect match with the miR-671 sequence. As a consequence, miR-671 can promote CDR1 linearization, resulting in its degradation [47]. Since the discovery of the relevance of circRNAs in regulating miRNA activity, several computational tools have been developed to help identify these molecules and to predict their potential interactions [48]. In parallel, the growing evidence surrounding miRNA sponges has motivated the development of several tools that are dedicated to identifying competing interactions between these different types of molecules [49].

The complex regulatory interactions between the different RNA molecules described above have been synthesized in the competitive endogenous RNA (ceRNA) hypothesis. This hypothesis states that mRNAs, lncRNAs (including circRNAs), and transcribed pseudogenes “communicate” with each other through MREs [50]. Unlike the traditional view, which states that MREs act exclusively as cis-regulatory elements that control translation and degradation dynamics, the ceRNA hypothesis proposes that the presence of these elements on a given RNA allows it to act as a trans regulator, indirectly influencing the expression of other mRNAs (Figure 1). This complex level of regulation can only be explored through the implementation of network-based approaches that integrate these multiple types of interactions as they occur within the cell. This implies defining coding and non-coding RNA species according to the specific combinations of MREs/miRNA sequences they present rather than according their gene ID and taking their relative abundances into consideration.

In summary, recent evidence shows that miRNA–mRNA target relationships can no longer be looked at from a simplistic perspective that only focuses on the binding of the seed sequence to a target mRNA and its resulting inhibition. The pool of transcripts that exist in a cell at a given moment contains players that are able to regulate each other by functionally modulating microRNA levels. The ceRNA hypothesis adds another layer of complexity to the miRNA–mRNA relationship and suggests that the actual ability of a given miRNA to act upon a target is dynamically influenced by the transcriptome environment it is in in addition to the relative affinity it displays to different targets.

## 5. Computational Approaches to Investigate miRNA–ceRNA Networks

After the proposal of the ceRNA hypothesis, several studies tried to contextualize its importance in different types of cancer and diseases (reviewed by [51]). Because this involves intricate relationships between different types of molecules, network approaches have been devised to better understand the underlying dynamics. The main focus of this section is to highlight the recent developments and achievements that have been made towards the process of building and analyzing ceRNA networks while also emphasizing how this work contributes to our ability to predict the functional impact of microRNAs in cell regulation.

Given the well-documented changes in ncRNA expression that occur in tumors, the investigation of ceRNA networks in the context of cancer has attracted significant interest. Shortly after the ceRNA hypothesis was put forth, Sumazin and colleagues set out to explore the possibility that the molecular species that regulate miRNA activity on their target RNAs without affecting miRNA expression levels could play a role in cancer [52]. Expanding on the original ceRNA hypothesis, they assumed that these modulators could be of two types: “sponge modulators”, which act by directly binding to the miRNA sequences, and “non-sponge modulators”. The latter represent either proteins or RNAs that act through a variety of alternative mechanisms, including those that precent the miRNA from binding to the target. To identify these regulators, the authors developed a new statistical method called Hermes, which was used to analyze matched glioblastoma mRNA and miRNA expression data from The Cancer Genome Atlas (TCGA) database and to identify potential miRNA/modulator/target triplets. To reduce the number of statistical tests performed by Hermes, the authors further developed a specific miR target-discovery algorithm called Cupid. This study uncovered a miR program-mediated posttranscriptional regulatory network of surprising magnitude, involving ~7000 genes acting as miR sponges and ~148 non-sponge regulators. The experimental validation of the results confirmed that this network regulates established tumor initiation drivers, such as the tumor suppressor PTEN. This study further suggested a possible mechanism for the observed loss of PTEN expression in a large number of gliomas with an intact PTEN locus. The Cupid method used in this study was further developed to support the simultaneous reconstruction of miRNA–target and ceRNA networks [53]. This approach builds on the predictions made by older algorithms [15,17,54], incorporating expression data and evidence of functional interactions to reduce false positive predictions while also identifying their mediated ceRNA interactions.

To try to uncover the emergent properties of ceRNA networks, several groups have applied mathematical modeling approaches. Bosia and colleagues developed a stochastic model to analyze the equilibrium and out-of-equilibrium properties of a network of M miRNAs interacting with N targets [55]. This model suggests that crosstalk can exist between ceRNA competing for the same miRs, but also that miRNAs can also crosstalk through common ceRNA molecules. To uncover the optimal conditions for ceRNA events to happen, Ala and colleagues [56] developed a mass-action model that was able to describe the molecular titration mechanism underlying the interaction between genes and a common set of miRs. Model predictions were experimentally validated by focusing on PTEN and its known ceRNA VAPA. This work highlighted the existence of integrated ceRNA and TF networks that can be significantly affected by ceRNA interactions under permissive molecular conditions. Finally, Figliuzzi and colleagues developed a minimal mathematical model to quantify the intensity of the interactions arising from ceRNA competition [57]. Their results suggest that both binding free energies and repression mechanisms are critical to induce ceRNA crosstalk. A later experimental systematic analysis of the target binding properties of the Argonaute–miRNA complex by Zamore’s lab raised significant questions as to whether these effects can be observed for the majority of miRNA–ceRNA pairs [58]. This study suggested that only a very limited number of miRNAs whose cellular concentration and target abundance are within a narrow range of values will be influenced by ceRNAs. Experimental studies from the Sharp and Stoffel labs performing a quantitative analysis of miRNA/ceRNA target pool levels and using reporter gene models to evaluate expression outputs supported the proposal that ceRNA regulation should only be significant under very specific conditions [59,60,61]. These studies suggest that the assumptions made by previous mathematical models were biased towards conditions that favor ceRNA regulation. To address the concerns raised by these studies, Chiu and colleagues [62] used an extended version of the Hermes algorithm to reverse engineer the ceRNA networks in prostate and breast adenocarcinoma using paired miRNA and mRNA expression profiles from the TCGA database. Their results suggest that a significant fraction of cancer genes may in fact be regulated by ceRNA interactions in the two tumor contexts. To further expand on this work, they generated a pan-cancer ceRNA interactome to analyze the overlap of the inferred interaction networks across several tumor contexts. The authors showed that ceRNA interactions that are mediated by many miRNAs are likely to have a context-independent physiological impact [63]. Their results suggest that a very large number of these interactions do exist, forming a unique regulatory network that may control gene expression across many cellular contexts [63]. A more recent study has tried to analyse the single-cell level response of varying miRNA concentrations using a combination of single-cell RNA-seq and mathematical modeling for the responses miRNA induction targets [64]. The results identified the key characteristics of miRNA–target regulations, pinpointing a specific range of parameters where ceRNA interactions may occur.

In spite of the controversy surrounding the degree to which ceRNA networks may impact cell regulation, this hypothesis has gained a lot of traction, and many research groups have developed computational approaches to uncover relevant interactions. These include CERNIA (ceRNA prediction algorithm), which considers validated miRNA–target interactions, non-canonical binding sites (5′ UTR and coding region), and tissue-specific characteristics [65]. SPONGE [66], which is based on a new mathematical approach, starts by identifying miRNA–target interactions from predictions and experimentally validated databases. It then applies sensitivity correlation coefficients that consider the role of more than one miRNA in a competing interaction between two target genes. Crinet (the ceRNA interaction network) aims to identify genome-wide ceRNA networks, trying to determine the drawbacks of existing methods [67]. Finally, LaceModule attempts to identify competing endogenous RNA modules based on the integration of the conventional Pearson correlation coefficient with a dynamic correlation measure called liquid association. This measure can incorporate the sensitivity of the correlation of ceRNA to miRs [68]. All of these methods have been devised to study and further expand our knowledge of the competing interactions that take place between different types of RNAs. The resulting predicted and/or validated ceRNA interactions are often curated and aggregated in public databases, making the information easily accessible to the research community (Table 1).

## 6. Conclusions and Future Perspectives

The last decade has revealed a previously unimagined layer of complexity at the post-transcriptional regulation level through the identification of novel types of cis and trans regulatory interactions established between miRNAs, mRNAs, and multiple classes of ncRNA molecules. Whether these mechanisms define pervasive regulatory networks that can profoundly influence functional outcomes or whether they are only relevant in very specific contexts remains a matter of debate. The ability to perform an increasingly quantitative characterization of the RNA sequence elements that are present within a single cell and to map protein outputs in these systems supported by on-going technological developments are expected to provide important contributions to our ability to address these questions. Given the complexity of the interactions that may be established, network-based analysis and mathematical models need to be developed to uncover not only the regulatory properties underlying this system, but, most of all, to achieve the holy grail of predicting the functional outcome of altering the abundance of specific players. For miRNA–ceRNA interactions, this is expected to require an improved ability to identify and quantify the sequence variants present in cell type-specific conditions, and furthermore, to efficiently integrate RRIs with PPIs to evaluate the final impact on cellular function.

Databases that are currently available, such as the TCGA, already provide access to large amounts of useful datasets that can be leveraged to understand ceRNA networks, namely matched miRNA and mRNA expression data in several types of cancer with adjacent normal tissue information. Regardless, the development of single-cell multi-omics methods will be of central importance for the fine dissection of these molecular processes. The development of computational methods to integrate global transcriptome interactions from multiple sources will further allow for a deeper understanding of how relevant these interactions are when defining cellular states. Moreover, it will also provide crucial information on how the local transcriptome environment influences the regulatory networks that are established.

## Figures and Tables

**Figure 1 biomolecules-12-00332-f001:**
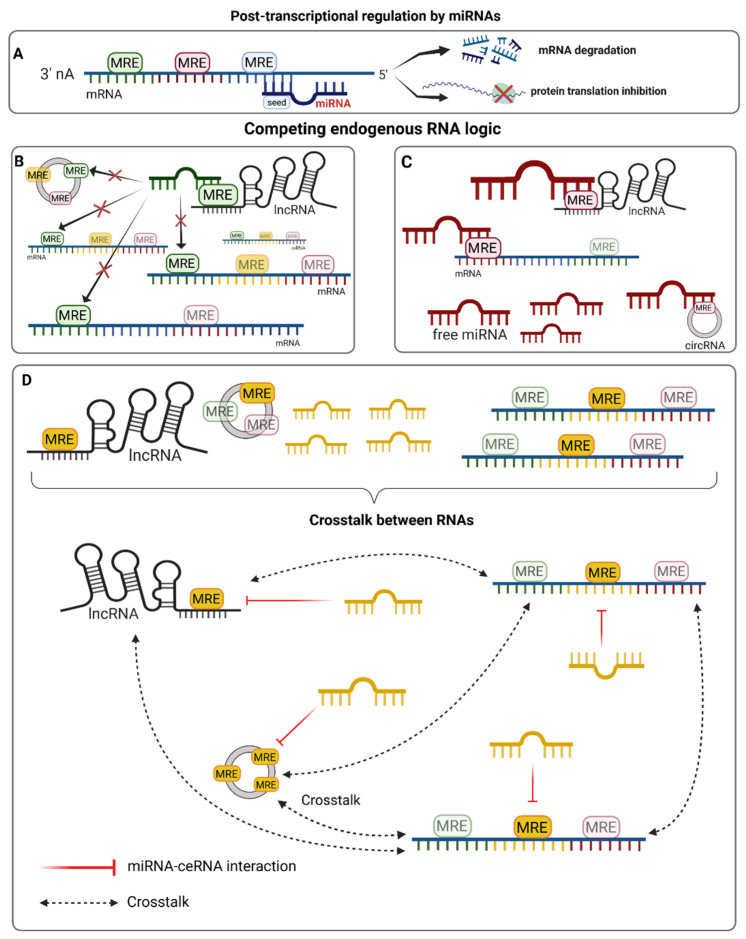
The competing endogenous RNA hypothesis. LncRNAs, circRNAs, and mRNAs can all co-exist in a given cellular context and form complex interaction networks. The type of molecules, their abundance, and presence and number of different microRNA responsive elements (MREs) with varying affinities for specific miRNA sequences can influence the way they interact and form competing endogenous RNA (ceRNA) networks. (**A**). The role of miRNAs in gene expression regulation is a well-studied mechanism that predominantly occurs through the binding of the seed sequence (nt 2–7) to a complementary MRE that is present in the 3′ UTR of mRNAs, leading to translational repression and/or degradation. However, other RNA molecules are able to interact with miRNAs and titrate their abundance, thus establishing a crosstalk interaction with other potential target molecules. Of note, these molecules often have MREs for many miRNAs, whereas each miRNA can usually target hundreds of different molecules. (**B**). In a situation where miRNAs are less abundant than a single ceRNA species, they may be sequestered and unable to interact with other targets. (**C**). When miRNAs are more abundant than the population of target RNA molecules, they are able to bind all of the targets, and some free miRNA may still exist in that context. (**D**). Complex ceRNA network interactions are more likely to happen when miRNAs and their corresponding MREs are present at similar concentrations. ceRNAs are able to influence the abundance of other RNA molecules that share the same MRE through the titration of free miRNAs. The amount of shared MREs directly influences the strength of crosstalk [50].

**Table 1 biomolecules-12-00332-t001:** Databases containing information on ceRNA interactions and networks.

Database	Link	Description	Reference
lncACTdb 3.0	bio-bigdata.hrbmu.edu.cn/LncACTdb	Experimentally supported ceRNA interactions and personalized networks	[69]
SomamiR	compbio.uthsc.edu/SomamiR/	Somatic mutations altering miRNA–ceRNA interactions	[70]
ENCORI	starbase.sysu.edu.cn/	Interactions for miRNA–mRNA, RBP–RNA and RNA–RNA	[71]
cerDB	oncomir.umn.edu/cefinder/basic_search.php	ceRNA–mRNA interactions	[72]
DIANA-LncBase v2	carolina.imis.athena-innovation.gr/diana_tools/web/index.php?r=lncbasev2	miRNA:lncRNA interactions that have been experimentally supported and in silico-predicted MREs on lncRNAs.	[73]

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
