# Peer review of "Network Approaches to Study Endogenous RNA Competition and Its Impact on Tissue-Specific microRNA Functions"

_biomolecules, 2022, doi:10.3390/biom12020332_

Round 1
Reviewer 1 Report
This review summarizes the state of the art of post transcriptional regulation and the significant role of miRNAs while interacting with other RNA species. It should be accepted as it is, however I recommend to read through once more for grammatical inconsistencies.
Line #249 givemn
Author Response
We would like to thank reviewer 1 for the critical reading of the manuscript and positive feed-back.
We have corrected the highlighted typo on line #249 (now line #342).
We have further carefully revised the manuscript for grammatical inconsistencies and introduced several corrections to make sentences clearer, including separating long sentences as requested by the second reviewer. In spite of a relatively large number of alterations, no change to overall content was made.
We believe that the manuscript is now improved and hope the reviewer agrees.
Reviewer 2 Report
The review by Marques and Gama-Carvalho describes the recent developments and achievements made towards the process of characterizing competing endogenous RNA (ceRNA) networks and their specific role in tissue-specific microRNA (miRNA) functions. They provide relevant information, logically structured, regarding the basics of these post-transcriptional regulatory mechanisms, which are becoming a rapidly growing field of interest.
I mostly found this review easy to read and included well-focused concise sections covering pertinent aspects of the topic, providing updated information from recent work on this topic. Further, this work is well supplemented with an illustrative and helpful figure, easy to understand.
I only have some general/minor comments that could help to get an improved version:
- I found the following paper relevant to be cited:
Ugo Ala; Competing Endogenous RNAs, Non-Coding RNAs and Diseases: An Intertwined Story; Cells, 2020, 9(7):1574
- I encourage the authors to break up those sentences that are split with several commas by rewriting them as two sentences (at times it is hard to keep track of the meaning as written).
Minor comments:
- Line 10: replace “a mRNA” for “mRNAs”
- Line 28: “Caenorhabditis elegans”?
- Line 215: Could the term “miRNA sponges” be further explained and referred to?
- Lines 393-401: Could these sentences become shorter for better understanding?
Author Response
We thank reviewer 2 for his critical reading of the manuscript, positive comments and suggestions for improvement.
We have tried to incorporate all the suggestions as follows:
Reviewer suggestion 1: "I found the following paper relevant to be cited: Ugo Ala; Competing Endogenous RNAs, Non-Coding RNAs and Diseases: An Intertwined Story; Cells, 2020, 9(7):1574"
We thank the reviewer for identifying this manuscript that had escaped us. We have introduced the citation at the beginning of the section on "Computational approaches to investigate miRNA-ceRNA networks", line 388, where we mentioned the relevance of these networks in the context of cancer and other diseases:
"After the proposal of the ceRNA hypothesis, several studies have tried to contextualize its importance in different types of cancer and diseases (reviewed by [51])."Reviewer suggestion 2: "I encourage the authors to break up those sentences that are split with several commas by rewriting them as two sentences (at times it is hard to keep track of the meaning as written)."
We have fully revised the manuscript text to correct for some gramatical inconsistencies (as highlighted by reviewer 1) and make sentences clearer, including breaking up long sentences into shorter ones. In spite of the number of changes introduced, care was taken not to alter the content/sense of the manuscript.
Minor comments:
"Line 10: replace “a mRNA” for “mRNAs” and Line 28: “Caenorhabditis elegans”?"
Changes made as requested.
"Line 215: Could the term “miRNA sponges” be further explained and referred to? "
Descriptions and citation of the term introduced (now line 291)
"Lines 393-401: Could these sentences become shorter for better understanding? "
The final paragraph of the manuscript, corresponding to lines 393-401 was revised and sentences were shortened.
We believe we were able to comply with all of the comments and suggestions made by the reviewer and thank him/her for the contribution to improve this manuscript.